# Stunting Was Associated with Reported Morbidity, Parental Education and Socioeconomic Status in 0.5–12-Year-Old Indonesian Children

**DOI:** 10.3390/ijerph17176204

**Published:** 2020-08-27

**Authors:** Moesijanti Y. E. Soekatri, Sandjaja Sandjaja, Ahmad Syauqy

**Affiliations:** 1The South East Asian Nutrition Survey (SEANUTS) Indonesian Team, Jakarta 12320, Indonesia; moesijanti@yahoo.com (M.Y.E.S.); san_gizi@yahoo.com (S.S.); 2Persatuan Ahli Gizi Indonesia, Jakarta 12320, Indonesia; 3Nutrition Department, Health Polytechnic, Ministry of Health of Jakarta II, Jakarta 12120, Indonesia; 4Department of Nutrition Science, Faculty of Medicine, Diponegoro University, Semarang 50275, Indonesia

**Keywords:** stunting, Indonesian children, morbidity, parental education, socioeconomic status

## Abstract

Stunting is highly prevalent in Indonesian children. The objective of this study was to identify the associations of stunting with morbidity, parental education and socioeconomic status (SES) in Indonesian children. The study population was part of the South East Asian Nutrition Surveys (SEANUTS). A total of 2236 Indonesian children aged 0.5 to 12 years, who had participated in the SEANUTS, were included in this study. Stunting was defined as height for age Z-score (HAZ) ≤ −2 using WHO criteria and severe stunting as HAZ ≤ −3. Information on morbidity, parental education and family SES were collected by structured questionnaires. ANOVA was used for evaluating differences across groups, with or without correction for confounders. The results showed that the overall prevalence of stunting was 31.4%. HAZ in stunted children was associated with disease incidence, including frequency, parental education and family income. There were no significant differences in HAZ values in stunted children with one or more bouts of infectious, digestive tract or respiratory tract illnesses compared to stunted children with no reported illness. The prevalence of stunting in Indonesian children was high and was strongly associated with child morbidity, parental education and SES.

## 1. Introduction

The Millennium Development Goals (MDG) report for Indonesia 2010 indicates that the prevalence of stunting in Indonesia is still high—between 30 and 40% [1]. Thus, stunting can be regarded as a public health problem [1]. Comparing stunting prevalence between the cross-sectional Indonesian basic health studies (Riskesdas) in 2018 (30.8%) [2] and 2013 (37.2%) [3] shows comparable values in stunting prevalence. However, Indonesia has seen a decline in childhood malnutrition from 1990 to 2013. The Indonesian basic health studies (Riskesdas) found a decreasing trend of stunting prevalence from 1992 to 2013—from 45.9% to 37.2% [2,4].

In Indonesia, the prevalence of stunting is much higher compared to neighboring countries such as Malaysia (8.4%) [5], Thailand (4.1 to 8.4%) [6] and Vietnam (14 to 15%) [7]. In the Philippines—one of the Association of Southeast Asian Nations (ASEAN) countries—the prevalence of stunting is about 30% in children under five years old [8].

Stunting is the consequence of chronic energy–protein malnutrition [9]. Moreover, other enhancing factors, including infections and diseases, are important in the etiology of stunting [8]. In a situation of insufficient supply of quality food, growth will be seriously hampered by repeated recovery periods from illnesses. In tropical countries, control of infectious diseases like malaria, dengue, or parasites play an important role, and it is partly a matter of hygiene [8]. As such, socioeconomic status (SES) is an important factor in the etiology of stunting. Not only will the lower SES have less access to quality food and food security [10], but the hygiene and infectious diseases are more likely to strike in poverty situations [11]. Accordingly, there is a need for an in-depth understanding of factors associated with stunting in Indonesia. The objective of this study, therefore, was to identify the associations of stunting (height for age Z-score (HAZ) ≤ −2) with morbidity, parental education and SES in Indonesian children.

## 2. Materials and Methods

The South East Asian Nutrition Survey (SEANUTS) was a multicenter nutrition study initiated by FrieslandCampina, The Netherlands and was conducted in Indonesia, Malaysia, Thailand and Vietnam in 2011 [12]. In Indonesia, the cross-sectional study comprised 7211 children aged 0.5–12.9 years, recruited using multistage cluster sampling, stratified for geographical location, sex and age. Details of the study design are published elsewhere [13]. Characteristics of the total population were published earlier [12]. The data in this report are only on the stunted children (*n* = 2236).

The study was conducted along the Guidelines of the Helsinki Declaration for experiments in humans and the study protocol was approved by the Committee of Health Research Ethics, the National Institute of Health Research and Development, the Ministry of Health, Republic of Indonesia, number LB.03.02/KE/6430/2010, and was also registered in the Netherlands Trial Registry number NTR 2462 [12]. Written informed consent was obtained from the parents prior to the study [12].

The length was measured supine in children younger than two years of age using a flat wooden measuring board. While in children older than two years old, standing height was measured using a wall-mounted stadiometer. Both measurements were done in duplicate with an accuracy of 0.1 cm. The average value was used in the calculations [14]. The WHO Child Growth Standards 2005 [15] and 2007 [16] were used to classify children as stunted (HAZ ≤ −2) or severely stunted (HAZ ≤ −3).

Information on reported morbidity, parental education, and SES was collected using standardized questionnaires. Information on diseases comprised of infectious diseases (malaria, measles, dengue and hepatitis), respiratory diseases (respiratory tract infection/RTI, pneumonia, tuberculosis and asthma) and digestive tract diseases (typhoid, diarrhea, oral problems). Parental education was categorized into six groups. Socioeconomic status was computed from valuable household possessions (e.g., house, cars, motorcycles, Internet, etc.) and monthly income expenditure, savings and jewelry; then, it was categorized into quintiles as very poor, poor, modal, rich and very rich [12].

All statistical analyses were performed on weighted data based on the population census 2011 using SPSS 24 (IBM Corp., Armonk, NY, USA) [12,17]. ANOVA was applied to test differences in HAZ across education levels, SES categories and groups of diseases corrected for age, area of residence and sex. Differences in the prevalence of stunting between areas of residence and sexes as well as across age groups were tested using chi-squared. Significance was set at *p* < 0.05.

## 3. Results

The results showed an overall prevalence of stunting of 31.4% with slight, but significant differences between boys and girls. Note the very much lower prevalence of stunting in the youngest age group compared to the older age groups and overall between urban and rural (Table 1). There were also significant differences between boys and girls in the percentages of stunted and severely stunted children in general and between rural and urban in boys. Comparing rural and urban in general, there was also a significant difference between stunted and severely stunted.

Table 2 shows that among stunting children (HAZ < −2 SD), HAZ values were significantly higher at higher educational levels of both father and mother. Furthermore, for SES, there was a tendency for higher HAZ values towards higher SES classes. However, there was a slight decrease in HAZ values in families with finished senior high school (SHS) and rich SES status.

Table 3 shows the most frequently reported illnesses in urban and rural areas and across SES classes. Only reported respiratory tract infection, asthma and oral problems were different between urban and rural areas (*p* < 0.05). However, for some illnesses, there were differences significantly (*p* < 0.05) across the SES groups, with generally lower illnesses in the higher SES groups.

Table 4 shows that the children with no illnesses in the previous month had higher HAZ values than those who reported one or more episodes of illness. There was a trend towards higher HAZ values if the number of reported illnesses was less. Digestive diseases were the most commonly reported illness suffered by the children over the last month and HAZ values were lowest in the children with repeated episodes of illnesses.

## 4. Discussion

In this study, the percentage of stunting in boys and girls differed slightly and higher in rural children than urban children. Differences in stunting prevalence between boys and girls have been reported in other studies. In China, the prevalence of stunting among school children aged 6–19 years old, although low, was significantly higher in girls (1.3%) than in boys (1.1%) [18]. A study in Bangladesh among young adolescents aged 10–17 years indicated that at the age of 12 years old, the prevalence of stunting was higher in girls (45.8%) than in boys (42.6%) [19]. In addition, differences in stunting prevalence between urban and rural children have been reported previously [20,21,22]. For example, a study in Iran in 841 children showed that stunting was significantly higher in rural than in urban children [23]. A review study in Indonesia indicated that children in rural areas were vulnerable to stunting than urban [24].

In the youngest age group stunting prevalence was much lower than in the older age groups (Table 1). These patterns were also found in studies in India and Guatemala [25]; for example, a recent study in Tangail District, Bangladesh, reported stunting among under-five-year-olds was much higher in rural (44%) than in urban children (3%) [20]. The lower prevalence of stunting at a young age suggests that frequent episodes of infectious diseases, with faltered growth as a consequence, are important in the etiology of stunting. In 2018, a basic health survey in Indonesia showed that 30.8% among under-five children, 23.6% in 5–12-year-olds, 25.7% in 13–15-year-olds and 26.9% in 16–18-year-olds were stunting [2]. This figure indicates that stunting in under five-year-old children are higher than the older children.

The results of this study show that in stunted children, the occurrence and frequency of illnesses in the previous month was associated with HAZ (Table 4). The results also showed that reported illnesses were generally higher in low SES children and rural children reported higher frequencies of illnesses. The relationship between poverty and undernutrition, including stunting, has been well reported [11,26]. Lower SES children are more likely to consume less and lower quality food [27] and also food diversity may be less [28], especially during the period when complementary feeding starts. Moreover, higher animal source food expenditure and plant source food expenditure decreased the risk of stunting in rural children and urban poor children 6–59 months old [24].

Another problem in relation to stunting is poor sanitation, which is more likely among families with low SES. Studies conducted in Iran and Bangladesh [23,28] indicated that limited access to community health services in remote areas with difficult or no access to clean water for drinking and bathing are most common in rural areas and these are enhancing risk factors for stunting [2,25]. On the contrary, the higher SES group more likely to have better food security in their families [10]. As a consequence, children could provide meals better in quantitively and qualitatively [10]. Moreover, a low-quality living environment, as often found in poor education families with low income, in addition to low-quality food and repeated episodes of illnesses may reduce immunity, which in turn may result in higher vulnerability—especially for diarrhea, respiratory diseases and infection diseases [29,30,31]. Malnutrition and/or repeated bouts of (infectious) diseases have a negative impact on growth, which explains why stunting is more common in rural areas and is often found to be higher at an older age [25].

The present study showed that a higher educational level coincided with a higher HAZ. This is also reported in other studies [29,32]. In developing countries such as Indonesia, the woman is key person to make an (informed) decision in the family when it comes to food and nutrition [33]. For example, in Zambia, in a study of 77 children aged 2 to 5 years old, 45 children were stunted [31]. More than half of these children were born from mothers who did not have any education, and about 30% of these children were from fathers with no formal education [31]. According to Semba et al. [33] parental education is a strong determinant of stunting in Indonesia. An increase in formal parental education will lead to a decrease of 3-5% in the risk of a child being stunting. In addition, from the present study, it is clear that parental education is related to the degree of being stunting. Those who finished at least elementary school had higher HAZ compared to those who did not finished elementary school (Table 2). This result was in line with a study using data in the multi-country analysis. They found that universal primary schooling would reduce stunting modestly (2.5%), but the mandatory middle schooling would reduce stunting 10%. This means that finished basic formal education of 9 years is important to combat stunting [34].

Furthermore, nutrition and health education programs for mothers of children and adolescence—especially adolescent girls—should include hygiene and sanitation to prevent infectious diseases. Nutrition education is one of the intervention programs categorized as a nutrition-specific program [35]. Pakistan and Vietnam were the countries where nutrition education put into account to prevent community health problems [36]. Other programs related to the income stimulation should focus on the very low SES, starting from training to make the people understand how to run the skill delivered then they continue step-by-step with their skill to produce their products to be sold in the market [37]. The fund should come from a non-governmental organization or the government. In Indonesia, every village was given funds to increase the well-being of the village community, improve the quality of human beings and reduce poverty at the village level [10].

The relationship between disease and stunting has been confirmed through many studies and is described as the ‘stunting syndrome cycle’ in children [38]. This syndrome describes that multiple pathologic changes reflected by short stature increases mortality and morbidity and may reduce physical and neuro-development [38]. In the current study, multiple reported illnesses also coincided with a lower HAZ value. It is understood that a metabolic consequence of diseases—especially infection—is an impaired absorption because of an altered gut lumen and mucosal injury, which in turn increases the risk of undernutrition or a worsening of an existing malnutrition [39]. This explains why multiple episodes of illnesses will impair growth and thus increase the risk of stunting. As such, programs aiming to eliminate stunting should ideally be combined with potable water safety and sanitation improvements, family planning and other factors that contribute to the risk of being stunted [33]. Nutrition education programs, focusing on exclusively breastfeeding, appropriate complementary feeding and adequate maternal and child care, should also be considered as the first 24 months of life. These are a crucial window of opportunities for reducing undernutrition [22,39]. Such programs should be implemented in order to correct the faltered growth especially in developing countries [9,33]. In addition, adequate access to health care is important to rural and urban communities because health service is one of the determinant factors related to stunting [24].

Some limitations in our study included that we did not measure other determinant factors related to stunting—especially maternal factors such as maternal age, parity, single mother, short stature, complications during pregnancy (e.g., hypertension, diabetes, anemia or poor weight gain), maternal employment and micronutrient supplementation. A previous study found that birth weight and length at birth as well as maternal short stature were the important risks factors of stunting in children aged 0–23 months in Indonesia [40]. Furthermore, child factors related to stunting such as preterm intrauterine growth restriction (IUGR), feeding practices (e.g., exclusive breastfeeding or formula, late or inadequate complementary feeding), parasitic infestation and AIDS and micronutrient supplementation (e.g., iron, vitamin D or vitamin A) were also not assessed.

## 5. Conclusions

In conclusions, our findings found that stunting is highly prevalent among Indonesian children. In stunted children, HAZ is significantly associated with a higher number of reported illnesses in the previous month, low parental education and low SES. The mandatory nine-year basic formal education program in Indonesia should be continued and should ideally also focus on nutrition and hygiene as factors related to illnesses. Nutrition and health education programs—especially for mothers of children under five—should including hygiene and sanitation. Income stimulation programs should focus on the very low SES.

## Figures and Tables

**Table 1 ijerph-17-06204-t001:** Prevalence of stunting by age group, sex and area of residence of Indonesian children.

Sex/Residence	Residence	Nutrition Status	Age Groups (Years)					
			0.5–0.9	1.0–2.9	3.0–5.9	6.0–8.9	9.0–12.9	All
			%	%	%	%	%	%
Boys ^1^	Urban ^2^	Severely stunted	3.8	6.5	3.8	1.7	2.7	3.4
		Stunted	10.1	18.8	21.0	19.7	18.4	19.2
	Rural ^2^	Severely stunted	4.6	15.5	16.9	7.7	10.6	11.8
		Stunted	6.9	32.3	28.6	29.7	31.9	29.3
	Total		12.7	36.4	35.0	29.6	31.3	31.7
Girls ^1^	Urban ^2^	Severely stunted	3.9	7.0	4.1	1.3	6.8	4.4
		Stunted	5.3	22.6	24.3	17.6	28.2	21.9
	Rural ^2^	Severely stunted	6.3	9.0	9.7	6.2	4.9	7.3
		Stunted	11.3	25.8	34.8	24.0	33.6	28.3
	Total		12.9	32.2	36.4	24.6	36.8	31.0
Urban ^2^		Severely stunted	3.9	6.7	4.0	1.5	4.6	3.9
		Stunted	7.8	20.7	22.6	18.6	23.2	20.5
	Total		11.7	27.4	26.6	20.1	27.9	24.4
Rural ^2^		Severely stunted	4.8	12.1	13.4	7.0	7.7	9.5
		Stunted	8.4	28.9	31.7	26.9	32.8	28.8
	Total		13.8	40.9	45.1	33.8	40.4	38.3

^1^ percentages were significantly different between the sexes (*p* < 0.05). ^2^ percentage significantly different between areas of residence (*p* < 0.05).

**Table 2 ijerph-17-06204-t002:** Relationship between parental education level, socioeconomic status and height for age Z-score (HAZ) in Indonesian stunted children (HAZ < −2 SD) ^1.^

Variables	HAZ	
	Mean	SE
Education level father		
No school ^a^	−2.81 ^d,e,f^	0.09
Not finished ES ^b^	−2.74 ^d,f^	0.04
Finished ES ^c^	−2.69 ^f^	0.02
Finished JHS ^d^	−2.59 ^a,b^	0.03
Finished SHS ^e^	−2.63 ^a^	0.03
Finished TE ^f^	−2.51 ^a–c^	0.07
Education level mother		
No school ^a^	−2.80 ^c–e,f^	0.07
Not finished ES ^b^	−2.78 ^d,e,f^	0.04
Finished ES ^c^	−2.68 ^a,c,f^	0.02
Finished JHS ^d^	−2.59 ^a,b,f^	0.03
Finished SHS ^e^	−2.61 ^a,b,f^	0.03
Finished TE ^f^	−2.41 ^a–e^	0.07
Socioeconomic status		
Very poor ^a^	−2.79 ^b–e^	0.02
Poor ^b^	−2.65 ^a,c,e^	0.02
Modal ^c^	−2.56 ^a,b^	0.03
Rich ^d^	−2,60 ^a,b^	0.03
Very rich ^e^	−2.54 ^a,b^	0.04

^1^ values corrected for sex, age and area of residence. ^a–f^ Different superscripts indicate significant differences across the education groups and SES groups. ES—elementary school; HAZ—height for age Z-score; JHS—junior high school; SHS—senior high school; TE—tertiary education.

**Table 3 ijerph-17-06204-t003:** Most frequent reported illnesses (%) in the previous month in stunted children by area of residence and socioeconomic status ^1^.

Illness	Residence		Socioeconomic Status				
	Urban	Rural	Very Poor ^a^	Poor ^b^	Modal ^c^	Rich ^d^	Very Rich ^e^
Respiratory							
ARI	43.3 ^2^	49.7	50.3 ^c–e^	49.3 ^d,e^	47.7 ^a,d,e^	42.1 ^a–c^	43.0 ^a–c^
Pneumonia	2.3	2.4	5.1 ^b–e^	1.3 ^a,d^	1.8 ^a,d^	2.3 ^a,b,e^	1.1 ^a,d^
Asthma	3.6 ^2^	5.4	5.6 ^c^	5.6 ^c^	3.2 ^a,b^	4.2	4.1
Tuberculosis	0.5 ^2^	0.9	0.5	0.7	1.0	0.5	1.0
Digestive							
Typhoid	3.7	4.0	5.2 ^d,e^	4.6 ^d,e^	4.6 ^d,e^	2.0 ^a–c^	2.8 ^a–c^
Diarrhea	13.2 ^2^	17.0	21.3 ^b–e^	15.9 ^a,c^	12.0 ^a,b^	13.1 ^a^	13.1 ^a^
Oral problems	40.1 ^2^	43.7	43.6 ^e^	41.0 ^d,e^	42.7 ^e^	45.2 ^d,e^	37.0 ^a–d^
Infectious							
Measles	5.0	4.7	5.7 ^e^	4.8 ^e^	5.2 ^e^	5.6 ^e^	3.0 ^a–d^
Malaria	0.3	0.3	0.6 ^b,d^	0.1 ^a^	0.2	0.1 ^a^	0.4
DHF	0.8 ^2^	0.4	0.5 ^d^	0.3 ^d^	0.2 ^d,e^	1.1 ^a–c^	0.8 ^d^
Hepatitis	0.4	0.2	0.3	0.5 ^c,e^	0.0 ^b^	0.4	0.2 ^b^

^1^ values corrected for age, sex and SES for ‘residence’ and for ‘SES’ corrected for age, sex and residence. ^2^ significantly different compared to rural. ^a–e^ Different superscripts indicate significant differences from SES groups. ARI—acute respiratory tract infection; DHF—dengue hemorrhagic fever.

**Table 4 ijerph-17-06204-t004:** HAZ among stunted children in relation with kind and frequency of reported illnesses in the previous month ^1^.

Kind of Illnesses	Number of Illnesses	HAZ	
		Mean	SE
Digestive illness ^2^	No reported illness	−2.66 ^a^	0.05
	One digestive illness	−2.80 ^b^	0.05
	At least two digestive illnesses	−2.78 ^b^	0.09
Respiratory illness ^3^	No reported illness	−2.67 ^a^	0.05
	At least one respiratory illness	−2.78 ^b^	0.04
Infectious illness ^4^	No reported illness	−2.73 ^a^	0.03
	At least one infectious illness	−2.72 ^a^	0.09
Total	No reported illness	−2.57 ^a^	0.07
	One illness	−2.70 ^b^	0.06
	Two illnesses	−2.78 ^b^	0.06
	At least three illnesses	−2.74 ^b^	0.07

^1^ values corrected for age, sex and area of residence. ^2^ digestive illness: typhoid, diarrhea or oral infection. ^3^ respiratory illness: acute respiratory tract infection (ARI), pneumonia, tuberculosis or asthma. ^4^ infectious illness: malaria, measles, dengue hemorrhagic fever or hepatitis. ^a–c^ Different superscripts indicate significant difference across the illness group. HAZ—height for age Z-score.

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
