# Peer review of "Stunting Was Associated with Reported Morbidity, Parental Education and Socioeconomic Status in 0.5–12-Year-Old Indonesian Children"

_ijerph, 2020, doi:10.3390/ijerph17176204_

Round 1

Reviewer 1 Report

The article is well designed and presented. However, the paper must refer to some of the latest references published from Indonesia as mentioned here (1-3). In addition to the given risk factors for child stunting “child morbidity, parental education and socioeconomic status, there are a number of other factors associated with child stunting in Indonesia that should also be mentioned/discussed together with the studied parameters. The conclusion also does not properly reflect the study results. The sentence mentioned in the conclusion “The mandatory 9-year-basic formal 184 education program in Indonesia should be continued and should ideally also focus on nutrition and 185 hygiene as factors related to illnesses” must be expressed/ discussed in the text, which is not the case here. Similarly, there is no mention about the income stimulation programs and their impact on the socioeconomic status (SES) and their association to child stunting. I have noticed a few typos/editorial corrections that should be taken care. There are some sentences that do not express the meaning properly and must be looked for clarity with moderate changes in English language. The manuscript is acceptable for publication with these corrections.

  1. Mahmudiono T, Andrias DR, Megatsari H, Nindya TS, Rosenkranz RR. Household Food Insecurity as a Predictor of Stunted Children and Overweight/Obese Mothers (SCOWT) in Urban Indonesia. Nutrients. 2018;10(5).
  2. Beal T, Tumilowicz A, Sutrisna A, Izwardy D, Neufeld LM. A review of child stunting determinants in Indonesia. Maternal & child nutrition. 2018;14(4):e12617-e.
  3. Utami N, Rachmalina R, Irawati A, Sari K, Rosha BC, Amaliah N, et al. Short birth length, low birth weight and maternal short stature are dominant risks of stunting among children aged 0-23 months: Evidence from Bogor longitudinal study on child growth and development, Indonesia. Malaysian journal of nutrition. 2018;24:11-23.

Reviewer 2 Report

Here the authors report the prevalence of stunting in Indonesian children and its association with various morbidities, parental education and social-economic status. I have the following comments for the authors.

Major:

  1. Authors should comment on the limitations of this study.
  2. While the reported factors (infectious, respiratory, and digestive co-morbidities, parental education, and social-economic status) are important in influencing shunting, authors should comment on why the following factors were not considered in this study?

Maternal factors – maternal age (teen pregnancy, or >35 years), parity (no. of children), single mother, short stature, complications during pregnancy such as hypertension, diabetes, anemia, poor weight gain, etc., maternal employment, micronutrient supplementation such as iron, and vitamin D

Child factors – preterm/IUGR, feeding practices (exclusive breastfeeding or formula, late or inadequate complementary feeding), parasitic infestation, and AIDS and micronutrient supplementation such as iron, vitamin D, and vitamin A.
